# *Pseudomonas aeruginosa pqs* Quorum Sensing Mediates Interaction with *Mycobacterium abscessus* In Vitro

**DOI:** 10.3390/microorganisms13010116

**Published:** 2025-01-08

**Authors:** Yun Long, Zhi Li, Menglu Li, Peiyi Lu, Yujia Deng, Pengyao Wu, Xue Li, Gangjian Qin, Jiamin Huang, Wenying Gao, Guobao Li, Tianyuan Jia, Liang Yang

**Affiliations:** 1Shenzhen Third People’s Hospital, National Clinical Research Centre for Infectious Disease, The Second Affiliated Hospital of Southern University of Science and Technology, Shenzhen 518112, China; longyunjn@163.com (Y.L.); wpykyzy@163.com (P.W.); 15041561203@163.com (X.L.); nira95@163.com (J.H.); jmjtgwy@126.com (W.G.); ligb2020@mail.sustech.edu.cn (G.L.); 2Department of Pharmacology, School of Medicine, Southern University of Science and Technology, Shenzhen 518055, China; 12233026@mail.sustech.edu.cn (Z.L.); 12133116@mail.sustech.edu.cn (M.L.); 12433125@mail.sustech.edu.cn (P.L.); 12111045@mail.sustech.edu.cn (Y.D.); qingj@sustech.edu.cn (G.Q.)

**Keywords:** *P. aeruginosa*, *M. abscessus*, PQS, membrane vesicles, the type VII secretion system

## Abstract

*Pseudomonas aeruginosa* and *Mycobacterium abscessus* are opportunistic pathogens that cause severe infections in hospitals, and their co-infections are increasingly reported. The interspecies interactions between these two bacterial species and their potential impacts on infections are largely unexplored. In this study, we first demonstrated that *P. aeruginosa* inhibits the growth of *M. abscessus* by iron chelating via *pqs* quorum sensing. Next, through proteomic analysis, we discovered that the PQS molecule significantly changed a large amount of protein expression in *M. abscessus*, including proteins involved in the type VII secretion system and iron homeostasis. Furthermore, we revealed that PQS significantly enhanced the production of bacterial membrane vesicles (MVs) by *M. abscessus*. Our study suggests that the *P. aeruginosa* PQS can serve as an interspecies signaling molecule to communicate with *Mycobacterium* and affect their physiology and virulence.

## 1. Introduction

*Pseudomonas aeruginosa* is a major opportunistic pathogen causing various human infections, including ventilator-associated pneumonia, burn wound infections, and chronic lung infections [1,2]. It is notorious for causing severe illnesses in immunocompromised individuals and cystic fibrosis patients [3]. *P. aeruginosa* is well known to form biofilms during infections, which further protect it from the host immune clearance and increase its resistance to antibiotics. The bacterium orchestrates virulence gene expression and group behaviors through a hierarchical quorum-sensing (QS) network, consisting of two Acyl homoserine lactone (AHL)-mediated systems and a *Pseudomonas* quinolone signal 2-heptyl-3-hydroxy-4-quinolone (PQS)-mediated system [4,5]. Most interestingly, PQS promotes *P. aeruginosa* to produce membrane vesicles (MVs), which belong to bacterial extracellular vesicles that have a profound impact on bacterial physiology and intercellular communication [6]. Disturbingly, certain *P. aeruginosa* strains have developed resistance to all available classes of antibiotics, highlighting the urgent need for further research on novel therapeutic strategies [7,8]. The increasing prevalence of multidrug-resistant strains poses a significant challenge to public health, necessitating the development of new antimicrobial agents and treatment approaches to combat this formidable pathogen effectively.

*Mycobacterium abscessus* (*M. abscessus*) is one of the most common non-tuberculous mycobacteria (NTM) pathogens that can cause opportunistic infections in clinical settings. Studies have demonstrated that *M. abscessus* can infect various anatomical sites, including the lungs, bones and joints, skin and soft tissues, and lymph nodes, with pulmonary infections being the most prevalent. The pathogen is frequently isolated from sputum samples or bronchial alveolar lavage fluid (BALF). Pulmonary infections caused by *M. abscessus* primarily occur in susceptible individuals, such as those with cystic fibrosis, chronic obstructive pulmonary disease (COPD), and bronchiectasis. These infections can lead to a rapid decline in respiratory function and are often recurrent. Despite ongoing efforts to combat *M. abscessus*-related diseases, progress remains slow, and current effective therapeutic strategies are still suboptimal.

Since *M. abscessus* and *P. aeruginosa* share the same ecological niche in the human lung [9], patients may be co-infected with both bacteria from environmental sources. Notably, 58–78% of *M. abscessus* infections are complicated by co-infection with *P. aeruginosa* [9]. This co-infection is especially common in cystic fibrosis (CF) patients and is clinically significant due to the increased antibiotic resistance of both pathogens [10]. In CF patients, the characteristic thick mucus impairs the lungs’ primary innate defense mechanism—the mucociliary clearance system [11]—, increasing the risk of infection by various pathogens, including *P. aeruginosa*, *Staphylococcus aureus*, and non-tuberculous mycobacteria (NTM). These pathogens are associated with severe pulmonary damage and chronic infections [12]. *P. aeruginosa* forms biofilms on the thickened, dehydrated mucus surfaces in CF patients, creating an impermeable barrier to antibiotic penetration and the host’s immune effectors [13]. These biofilms contribute to chronic infections by enhancing antibiotic tolerance, resisting phagocytosis, and promoting immune-mediated inflammation [14]. Similarly, *M. abscessus* can also form biofilms, further complicating treatment. Co-infections with these pathogens are linked to increased morbidity and mortality, owing to prolonged treatment regimens, significant side effects, multidrug resistance, and poor success rates in clearing respiratory tract infections.

Among all the diseases caused by *Mycobacterium*, tuberculosis (TB) is a severe global disease that significantly impacts the daily life of a large population globally [15]. This disease primarily affects the lungs, resulting in long-term complications such as bronchovascular distortions, bronchiectasis, emphysema, and fibrosis [16]. Structural and immunological changes in the lungs also make patients more susceptible to other pathogenic microorganisms. Liang et al. [17] have highlighted a notable rise in infections caused by NTM (non-tuberculous mycobacteria) among TB patients, including the rapid-growing *M. abscessus* [17]. TB is a classic zoonotic disease caused by various strains within the *Mycobacterium tuberculosis* complex (MTBC) [18,19]. Among these, *Mycobacterium tuberculosis* (Mtb) is the primary pathogen responsible for human tuberculosis, while *Mycobacterium bovis* predominantly causes bovine tuberculosis [20]. However, *M. bovis* can also infect humans, resulting in zoonotic tuberculosis [20]. *M. abscessus* infections are linked to higher morbidity and mortality due to prolonged treatment, significant side effects, and antimicrobial resistance [21]. Similar to *P. aeruginosa*, *M. abscessus* also produces MVs [22]. Notably, some TB patients are co-infected with both *P. aeruginosa* and *M. abscessus* [9]. The potential interacting mechanisms between *P. aeruginosa* and *M. abscessus* in tuberculosis patients have not been reported to date.

Despite the various medical challenges outlined above, research on the interactions between *P. aeruginosa* and *M. abscessus* remains relatively scarce. Therefore, it is crucial to further elucidate and characterize the mechanisms underlying co-infection and interactions between these two bacterial species. In this study, a total of 770 tuberculosis patients were enrolled to identify different microbial co-infection patterns and several cases of *P. aeruginosa*-*M. abscessus* were observed. We next found that *P. aeruginosa* is able to inhibit *M. abscessus* growth and induce its MV formation via PQS. Additionally, proteomic profiling was employed to examine the influence of PQS on *M. abscessus*, aiming to elucidate the impact of *P. aeruginosa* on *M. abscessus* virulence.

## 2. Material and Methods

### 2.1. Bacterial Strains and Growth Conditions

The *M. abscessus* ATCC19977 reference strain was sourced from the Guangdong Microbial Culture Collection Center (GDMCC). Clinical *M. abscessus* strains were obtained from Shenzhen Third People’s Hospital’s Laboratory Medicine Department. These isolates were derived from respiratory samples of tuberculosis patients co-infected with *P. aeruginosa* and *M. abscessus* (n = 5). To study colony morphotypes, the strains were grown on Middlebrook 7H10 agar with 10% OADC (oleic acid, albumin, dextrose-catalase) enrichment (Becton Dickinson, Heidelberg, Germany) and 0.5% glycerol at 37 °C for 3–4 days. In liquid culture, the strains were grown in Middlebrook 7H9 broth (M7H9) containing 10% ADC (Becton Dickinson), 0.05% Tween-80, and 0.5% glycerol. Growth curves were assessed using a Tecan Spark plate reader (OD600) (Tecan Trading AG, Männedorf, Switzerland) over 48 h after overnight subcultures were transferred to 96-well plates. To study the effects of PQS on *M. abscessus* 19977, PQS (30 μM) was added to PQS-free M7H9 medium for 12 h. PQS (2-Heptyl-3-hydroxy-4(1H)-quinolone) was purchased from Sigma (St. Louis, MO, USA, Cat#94398).

### 2.2. P. aeruginosa Mutant Construction

*P. aeruginosa* Δ*pqsR* mutant strain was generated through homologous recombination, following the protocol by Choi and Schweizer [23]. The targeting fragment was designed via overlapping PCR to include a gentamicin-resistance cassette, and the resulting fragments were inserted into the pK18 suicide vector to create gene knockout plasmids. These plasmids were transformed into *E. coli* strain RK600 and transferred to PAO1 through conjugation. *P. aeruginosa* PAO1 was used as a model organism for all experiments. Integrants were selected on agar plates containing 30 µg/mL gentamicin (Gm30). To resolve merodiploids, Gm-resistant colonies were streaked on LB Gm30 plates with 5% sucrose to isolate single colonies. Mutants were screened by PCR using flanking primers, which were confirmed by Sanger sequencing.

### 2.3. Proteomics Analysis

Proteins were extracted from *M. abscessus* using a bead homogenizer in lysis buffer with protease and phosphatase inhibitors. The proteins were quantified by BCA assay and qualified by SDS-PAGE [24]. Each qualified protein sample (100 μg) was diluted with 50 mM NH_4_HCO_3_ to four times its volume. Then, 2.5 μg of trypsin enzyme was added and incubated at 37 °C for 4 h. The resulting peptides were desalted with a Strata X column, dried, and freeze-dried. The peptides were reconstituted in mobile phase A (2% ACN, 0.1% FA), centrifuged at 20,000× *g* for 10 min, and the supernatant was used for injection. Separation was performed using a Thermo UltiMate 3000 UHPLC (USA). For DIA analysis, the peptides were ionized by nanoESI and injected into a Q-Exactive HF X mass spectrometer (Thermo Fisher Scientific, Bremen, Germany) in DIA mode. The data were analyzed using the mProphet algorithm for quality control and iRT peptides for retention time calibration. Using the SWATH-MS target-decoy model, false positives were controlled with a 1% FDR, allowing for the identification and quantification of peptides and proteins. The R package MSstats, version 4.0 was used to evaluate significant differences in proteins or peptides across groups, with a fold change > 2 and *p* value < 0.05 as criteria. Functional annotation and enrichment analysis were then performed on the differential proteins.

### 2.4. Extraction and Purification of M. abscessus MVs

Bacterial strains were cultured in M7H9 medium until reaching the exponential phase, then harvested by centrifugation at 8000× *g* for 15 min at 4 °C. Then, the supernatant was filtered through a 0.45 µm filter (Biofil, Guangzhou, China) and concentrated using 100 KD Vivaflow membranes (Sartorius AG, Göttingen, Germany). The concentrated supernatant was centrifuged at 50,000× *g* for 2 h at 4 °C to pellet crude MVs. These MVs were further purified by density gradient centrifugation (15–60%) at 100,000× *g* with Opti-Prep Gradient Medium (Sigma Aldrich, St. Louis, MO, USA) for 16 h at 4 °C. The 35–45% fraction containing purified MVs was washed in PBS for 1.5 h at 4 °C, and the final pellets were resuspended in PBS.

### 2.5. Characterization of MVs by Transmission Electron Microscopy (TEM) and Nanoparticle Tracking Analysis (NTA)

Purified MVs were visualized by negative staining TEM. Briefly, 5 μL of MVs were incubated for 1 min on a glow-discharged 200-mesh carbon grid (Beijing Zhongjingkeyi Technology, Beijing, China) and contrasted with phosphotungstic acid. The grids were then viewed on an HT7700 transmission electron microscope (Hitachi, Tokyo, Japan) at 100 kV with a high-sensitivity CCD camera. For NTA, MV quantification and size characterization were performed on a NanoSight NS300 (Malvern Instruments Ltd., Malvern, UK) using a 448 nm laser in scatter mode. MV samples were diluted 500× in PBS to achieve a concentration of 1–10 × 10^8^ particles/mL, injected with a 1 mL syringe, and videos were captured in triplicate for 30 s. The mean size and concentration values were analyzed using NanoSight software (version 3.0). For northern blotting, external RNAs were removed by incubating samples with 10 ng/μL RNase A/T1 (Thermo Scientific, Waltham, MA, USA) for 30 min at 37 °C, diluting 1/100 with water, and pelleting at 50,000× *g* for 1 h.

### 2.6. Scanning Electronic Microscopy (SEM)

After 12 h of growth in the PQS-supplemented M7H9 medium, *M. abscessus* samples were fixed overnight in 2.5% glutaraldehyde. The samples were washed three times with 0.1 M phosphate buffer (PB, pH 7.4) for 15 min each, followed by fixation with 1% osmium tetroxide at room temperature, protected from light, for 1–2 h. Afterward, the samples were washed again with 0.1 M PB three times, for 15 min each. Dehydration was carried out by immersing the samples sequentially in ethanol solutions of increasing concentrations (30%, 50%, 70%, 80%, 90%, 95%, and twice in 100%) for 15 min at each step, followed by treatment with isoamyl acetate for 15 min. The samples were then dried using a critical point dryer, attached to conductive carbon tape, and sputter-coated with gold for about 30 s. Finally, the samples were observed and imaged under a scanning electron microscope (HITACHI).

### 2.7. qRT-PCT Analysis

Messenger RNA level was analyzed by quantitative real-time PCR (qRT-PCR) using BioRad iQ5 Real-Time PCR Detection System (Hercules, CA, USA). The primer sequences used in this study are listed in Table 1. Relative quantitative real-time PCR was conducted by SYBR Green Master Mix (TaKaRa, Kusatsu shi, Japan) on cDNA generated from the reverse transcription of the purified RNA. After preamplification (95 °C for 2 min), the PCRs were amplified for 50 cycles (95 °C for 15 s and 60 °C for 1 min) on the iQ5 real-time PCR detection system. Each mRNA expression was normalized against *sigA* mRNA expression using the comparative cycle threshold method [25,26]. The relative expressions of iron-regulation-related genes in *M. abscessus*—*ideR*, *bfrB*, *mbtE*, *fecB*, *irtA*, and *aqdC*—were tested by qRT-PCR in this study. Additionally, we evaluated the expression levels of some genes associated with the type VII secretion system, including *MAB_3747c*, *MAB_4316*, *esxT*, *esxU*, *esxH*, *esxG*, and *sigM* (Table 1).

### 2.8. Ethics Statement

All the samples (bacterial isolates) were retrieved from the Laboratory Medicine of Shenzhen Third People’s Hospital. All the samples used in this study were anonymized. This study and the use of the samples were approved by the local ethics committee of Shenzhen Third People’s Hospital (Internal Study No. 2022-052-02).

### 2.9. Statistical Analysis

Statistical analyses were conducted using GraphPad Prism software, version 9.0. Data were analyzed with a two-sided Mann–Whitney U test, and *p* values were corrected using the Benjamini–Hochberg method. The results represent the means ± SD of three independent experiments, unless specified otherwise. *p* values < 0.05 were considered statistically significant. False discovery rate controls were set at 10% (*q* < 0.1) using the Benjamini–Hochberg procedure. Details for all the statistical tests can be found in the figure legends.

## 3. Results

### 3.1. P. aeruginosa Inhibited the Growth of M. abscessus

In this study, we conducted a retrospective analysis of clinical patients infected with *M. abscessus* (Mab group), *P. aeruginosa* (Pa group), and co-infected with both among tuberculosis patients in Shenzhen Third People’s Hospital over three years, as illustrated in Appendix A. The results showed that, in addition to tuberculosis infection, 712 patients were additionally infected with *M. abscessus*, 53 patients were infected with *P. aeruginosa*, and 5 patients were infected by both species. To investigate the interaction between *P. aeruginosa* and *M. abscessus*, the *P. aeruginosa* PAO1 wild-type strain and *M. abscessus* 19977 wild-type strain were independently inoculated on a plate to observe their interactions at varying distances. We performed a series of pairwise proximity assays to explore the extracellular-product-mediated interactions between *M. abscessus* and *P. aeruginosa*. As shown in Figure 1A, we observed the interaction between *M. abscessus* and *P. aeruginosa* at different distances (15 mm, 20 mm, 25 mm, 30 mm), using the center of the colony on the first day as the reference point. Next, we measured the growth distance of *M. abscessus* to the left and right at different distances and time intervals. We then calculated the fold change in the ratio of the leftward to rightward growth distances (Figure 1B). The results revealed that as *M. abscessus* and *P. aeruginosa* grew, the closer the distance between them, the greater the displacement of *M. abscessus* appeared (Figure 1C). These findings indicate that *P. aeruginosa* can inhibit the growth of *M. abscessus*, providing a basis for further research into bacterial antagonism and virulence mechanism.

In addition, we observed that the red coloration around *P. aeruginosa* diminished as the distance between *P. aeruginosa* and *M. abscessus* decreased. Therefore, we hypothesized that *P. aeruginosa* inhibits the growth of *M. abscessus* by competing for iron. PQS is a molecule produced by *P. aeruginosa* for iron chelation, which is crucial for *P. aeruginosa*’s iron acquisition. To test this hypothesis, we constructed a PQS defective Δ*pqsR* strain of *P. aeruginosa* and conducted co-culture experiments with *M. abscessus* (Appendix A). The results showed that, compared to the wild-type *P. aeruginosa* PAO1, the *P. aeruginosa* Δ*pqsR* mutant had a reduced inhibitory effect on the growth of *M. abscessus*, as evidenced by the smaller displacement of *M. abscessus* compared to that in Figure 1. These findings indicate that *P. aeruginosa* can influence the growth of *M. abscessus* through PQS.

### 3.2. Global Impact of PQS on M. abscessus

PQS is an important quorum-sensing molecule secreted explicitly by *P. aeruginosa*, known for regulating various virulence factors [4]. To investigate the influence of PQS on *M. abscessus*, proteomic analysis was used to identify differentially expressed proteins (DEPs) between the *M. abscessus* (Mab) and Mab + PQS groups. The number of differentially expressed proteins in the PQS-treated groups was identified, and most of them were upregulated (Figure 2A). The information on the specific proteins is shown in Appendix A. A heatmap of the proteomic analysis for 45 highly differentially expressed proteins from these groups was generated (Figure 2A). The top five differentially expressed proteins are indicated in the volcano plot (Figure 2B). The information on the specific proteins is shown in Table 2. Among the differentially expressed proteins, the WXG100 family type VII secretion target (BIMJN1) was significantly upregulated in the Mab + PQS group compared to the Mab group, indicating that PQS activates the type VII secretion system (T7SS) in *M. abscessus*.

To further investigate the influences of PQS on *M. abscessus*, GO and KEGG enrichment analyses of proteomics profiling results were performed on the Mab and Mab + PQS groups. The GO results from the proteomics data revealed significant changes in the Mab + PQS group in the GO terms “4 iron, 4 sulfur cluster binding”, “heme binding”, and “metal ion binding”, which are involved in regulating the iron synthesis, thereby ensuring that iron is utilized by the cells in a functional form (Figure 2C). Additionally, the GO terms “cytolysis in other organisms” and “extracellular region” exhibited significant changes in the Mab + PQS group. These terms are involved in regulating bacterial pathogenicity, which in turn activates the host’s immune response. In the KEGG analysis of proteomics, type I polyketide structures, biosynthesis of antibiotics, and biosynthesis of secondary metabolites pathways were shown to be highly influenced by PQS (Figure 2D). These findings indicate that PQS has a great global impact on *M. abscessus*, which may activate the type VII secretion system and alter the iron homeostasis of *M. abscessus*.

### 3.3. PQS Promotes the Production of MVs by M. abscessus

PQS could activate bacterial MV productions, including Gram-negative *E. coli* K12 and Gram-positive *Bacillus subtilis* 168 [27]. PQS promotes MV formations through binding to LPS and induces membrane curvature [28]. To examine whether PQS activates the production of MVs in *M. abscessus* with thick and unique cells wall but without LPS, *M. abscessus* was cultivated in medium with and without the addition of PQS, as shown in Figure 3A. The addition of PQS to the culturing medium led to a red color, indicating that PQS captured iron ions in the medium. The growth curve over time was measured, with Mab + DMSO serving as a positive control. As shown in Figure 3B, the growth trends of the bacteria were similar across all treatment groups, indicating that PQS and DMSO did not significantly affect bacterial growth. As shown in Figure 3C, the PQS addition tube shows a noticeable precipitate in the red box region, indicating a higher concentration of MVs. To verify these results, the protein concentration in the precipitates from both groups was measured and shown in Figure 3D. The results demonstrated that the protein concentration in the Mab + PQS group (58.6 ng/μL) was significantly higher than in the Mab group (7.9 ng/μL), indicating that more MVs existed in the Mab + PQS group.

MVs were isolated from the culture supernatants of *M. abscessus* grown in the presence or absence of PQS, as previously described. The characteristics of *M. abscessus* MVs exposed to PQS were further detected through electron microscopy imaging (Figure 3E). It can be observed that MVs from the Mab group are sparsely distributed and present in lower quantities, whereas MVs from the Mab + PQS group are significantly increased in number. Further nanoparticle tracking analysis (NTA) was conducted on the MV concentrations of the Mab and Mab + PQS groups (Figure 3F), revealing a significantly higher MV concentration in the Mab + PQS group compared to the Mab group, which indicates that PQS enhances the production of *M. abscessus* MVs. NTA was utilized for further analysis of MV size distribution in both the Mab and Mab + PQS groups. Notably, the results demonstrate that MVs from the Mab group exhibit a relatively concentrated size distribution, primarily around 100 nm, while MVs from the Mab + PQS group also exhibit a concentration around 100 nm (Figure 3G). These results revealed that PQS significantly increased the MV productions in *M. abscessus*.

### 3.4. PQS Exposure Changes Cell Morphology of M. abscessus

To further investigate the influence of PQS on the secretion of MVs from *M. abscessus*, we examined the *M. abscessus* cell morphology under two conditions using both scanning electron microscopy (SEM) and transmission electron microscopy (TEM). As shown in Figure 4A, the SEM results revealed that MVs were released from the surface of *M. abscessus*, with the number of MVs on individual cells in the Mab + PQS group being significantly higher compared to the Mab group. Additionally, PQS exposure led to a morphological change in *M. abscessus*, with cells exhibiting a wrinkled surface appearance. Subsequently, negative staining was used to observe the MVs around the *M. abscessus* cells, confirming that the Mab + PQS group had a small number of MVs, while the Mab group had almost none (Figure 4B). These findings indicate that PQS enhances the secretion of MVs from *M. abscessus* and alters its morphology.

### 3.5. Gene Expression Analysis upon PQS Exposure in M. abscessus

Iron is a vital nutrient involved in a wide range of enzymatic functions and biological processes. It is essential for bacterial growth and virulence. PQS, produced within the cell, is secreted into the extracellular space by *P. aeruginosa* and then integrated into the outer membrane to participate in the formation of outer membrane vesicles (OMVs). Under low-iron conditions, the PQS in OMVs forms a PQS-Fe^3+^ complex with extracellular Fe^3+^ [5]. In our proteomic analysis, we detected changes in iron-related proteins, such as B1MEZ2 (assimilatory sulfite reductase [ferredoxin]) and B1MP80 (probable ferredoxin). The qRT-PCR analysis also revealed that the relative expressions of iron-regulation-related genes—*ideR*, *bfrB*, *fecB*, *irtA*, and *aqdC*—were significantly upregulated in PQS-treated *M. abscessus* compared to the control group (*p* < 0.05) at 12 h (Figure 5A).

The ESX-3 type VII protein secretion system in *M. abscessus* plays a crucial role in the host’s inflammatory and pathological responses during infection. Studies have reported a correlation between ESX-3 and an iron uptake system [29]. In our previous proteomic analysis, we also detected an upregulation of proteins related to the type VII secretion system, such as B1MJN1 (WXG100 family type VII secretion target) and B1MG76 (ESX-1 secretion-associated protein). To validate these findings, we assessed the expression levels of the corresponding genes for B1MJN1 and B1MG76 (Figure 5B) and found that *MAB_3747c* and *MAB_4316* were indeed upregulated in PQS-treated *M. abscessus*. Additionally, we evaluated the expression levels of other substrate genes associated with the type VII secretion system, including *esxT*, *esxU*, *esxH*, *esxG*, and *sigM* (Figure 5B). The results showed varying degrees of upregulation in these genes as well. Collectively, these findings demonstrate that PQS promotes the activation of the type VII secretion system and induces iron starvation in *M. abscessus*.

## 4. Discussion

Despite the existing literature reporting the coexistence of *M. abscessus* and *P. aeruginosa* in some pulmonary infections, such as cystic fibrosis (CF) [30], there is still limited understanding of the molecular interaction between these two opportunistic pathogens. This study shed light on the complex interactions between *M. abscessus* and *P. aeruginosa* in vitro, providing insights for future research into bacterial infections and potential therapeutic targets.

Until now, only a few studies [10,31] have reported research on concurrent infections of *P. aeruginosa* and *M. abscessus*. For instance, Hsieh, M.H., et al. [31] have observed that NTM isolates obtained from the sputum of non-CF bronchiectasis patients are linked to more pronounced declines in forced expiratory volume in one second (FEV1) and increased frequency of acute exacerbations (AE). Individuals harboring both positive NTM and *P. aeruginosa* isolates concurrently exhibit the most significant deterioration in pulmonary function and experience the highest rate of acute exacerbations. Rodriguez-Sevilla, G. et al. [9] have revealed that antibiotic treatment targeting *P. aeruginosa* in CF patients may result in enhanced viability of *M. abscessus* within dual-species biofilms, potentially facilitating the onset of NTM pulmonary disease. However, previous studies have primarily focused on how co-infections exacerbate the disease characteristics in patients, and current research on the mechanism of *P. aeruginosa*’s impact on *M. abscessus* remains quite limited. *P. aeruginosa* is typically an extracellular bacterium. Nevertheless, under conditions such as weakened host immune systems, tissue damage, or the use of immunosuppressive agents, *P. aeruginosa* is more likely to survive within macrophages [12,32]. The interaction between *P. aeruginosa* and *M. abscessus* may be attributed to nutrient deprivation, compromised immunity, and lung tissue damage in individuals affected by tuberculosis. Consequently, we conducted in vitro studies to investigate whether PQS can interact with the cell wall of *M. abscessus* and to explore the potential effects of this interaction on *M. abscessus*.

PQS is the autoinductor of *pqs* QS systems in *P. aeruginosa*, influencing the regulation of various virulence factors, iron acquisition, and biofilm formations [33,34,35,36]. The activation of PQS and other QS pathways results in the synthesis of various virulence factors, facilitating *P. aeruginosa*’s adaptation and survival in polymicrobial environments, including against *Staphylococcus aureus* [37,38]. MVs serve as vehicles for the delivery of toxins, enzymes, and other bioactive molecules such as DNA/RNA [39], which can impact host cells by eliciting immune responses or causing infection [40]. PQS could activate many bacterial MV productions, such as Gram-negative *E. coli* K12 and Gram-positive *Bacillus subtilis* 168 [27]. PQS promotes MV formations through binding to LPS and induces membrane curvature [28]. It is worth pointing out that *M. abscessus* belongs to the non-tuberculous mycobacteria with thick and unique cell wall and without LPS [41]. However, until now, few studies have reported the relationship between PQS and *M. abscessus* and whether PQS could also increase *M. abscessus* MV production is unclear. It is worth pointing out that, with proteomic results and PCR results, this study confirmed that PQS promotes iron acquisition in *M. abscessus*. Some genes related to iron acquisition were tested by PCR in this study. For instance, bacterioferritin stores excess iron to prevent iron toxicity and releases stored iron under conditions of iron deficiency. Mtb encodes bacterioferritin BfrB (Rv3841), while homologous proteins (MAB_0126c and MAB_0127c) are predicted in *M. abscessus*. In mycobacteria, these ferritins are regulated by IdeR (iron-dependent regulatory factor) in response to varying environmental iron levels [42]. Mtb acquires iron through siderophores, including mycobactins (MBTs) and carboxymycobactins (cMBTs), synthesized by the mbt gene cluster. In *M. abscessus*, the homolog of the core gene *mbtE* is *MAB_2248c* [43]. The transporter IrtA, forming a complex with IrtB, mediates the transport of iron-cMBT complexes across the inner membrane [44]. Periplasmic binding proteins (PBPs) also play a critical role in iron-cMBT uptake. Ligand-binding assays have shown that the PBP FecB binds both iron-cMBT and heme [45]. Previous studies [46] have reported that *M. abscessus* can degrade exogenously supplied PQS. AqdC, a PQS dioxygenase that catalyzes quinolone ring cleavage, is the key enzyme in this degradation pathway. Iron sequestration by host proteins is a defense mechanism against bacterial pathogens, which require iron for their metabolism and virulence [47]. Lin et al. [47] reported that *P. aeruginosa* recruits PQS-containing outer membrane vesicles for iron acquisition. They also suggested that it would be interesting to investigate whether the PE–PPE proteins are involved in the recruitment of MVs for iron acquisition by directly interacting with mycobactin [47]. Therefore, iron acquisition is closely linked to the secretion of microbial MVs. In this study, experimental validation demonstrated that PQS significantly increases the production of MVs in *M. abscessus*, without affecting the size of the MVs. This study supports previous studies showing that PQS promotes the production of MVs in the *Mycobacterium*. Moreover, we have unveiled for the first time the impact of PQS on the production of MVs by *M. abscessus* in vitro.

Interestingly, we also first identified the protein of the WXG100 family type VII secretion target, which could be regulated by PQS in *M. abscessus*. The type VII secretion system (T7SS), also known as the early secretory antigenic target (ESX) system, is highly conserved among mycobacterial species and is essential for protein secretion. To date, five ESX systems (ESX-1 to ESX-5) have been identified in mycobacteria. In *M. abscessus*, T7SS includes two gene clusters, ESX-3 and ESX-4 [48]. EsxG and EsxH are substrates of the ESX-3 system, while EsxU and EsxT are substrates encoded by the ESX-4 cluster [29,49]. SigM, a transcription factor, has been reported to induce the expression of ESX-4 substrates [50]. Through the T7SS system, bacteria can encapsulate certain proteins, toxins, and other biomolecules within MVs, thereby releasing these substances to the external environment. Lin et al. revealed that the Esx-3 type VII secretion system also participates in the assimilation of mycobactin-bound iron by secreting a pair of proteins belonging to the PE–PPE family (PE5–PPE4) [47]. Additionally, Bythrow et al. [29] reported that the type VII ESX-3 system is closely related to the iron uptake mechanism in *M. abscessus*. These findings suggest a connection between the type VII secretion system and iron metabolism. However, whether the type VII secretion system is directly related to the secretion of MVs in *M. abscessus* and the underlying mechanism remains unclear. We propose that the type VII secretion system influences iron acquisition, thereby affecting iron utilization in *M. abscessus*, which in turn promotes the secretion of more MVs. However, the specific mechanism needs to be further validated in future studies.

Currently, there is an increasing incidence of tuberculosis patients co-infected with *P. aeruginosa*. This may be related to the ability of *P. aeruginosa* to colonize a wide range of ecological niches, such as air pollutants, animal hosts, and humans [2,51]. Mtb MVs have been found to trigger various immune responses, including inflammation and antigen presentation, playing a crucial role in immune modulation [52,53]. Moreover, these MVs are closely linked to intercellular communication. To date, the impact of *P. aeruginosa* on *M. tuberculosis* exosomes MVs remains unexplored. Our research provides new insights into addressing the co-infection of *P. aeruginosa* in vitro. It is important to note that in countries with a high burden of tuberculosis, the coexistence of bovine and human tuberculosis poses a significant challenge, particularly due to the ability of cattle to harbor both *M. bovis* and *M. tuberculosis* [54]. This dual carriage represents a substantial threat to the effectiveness of the human “End TB Strategy” and bovine tuberculosis eradication programs [55]. This study also may provide new perspectives for addressing zoonotic tuberculosis and hold promise for the development of novel diagnostic methods.

However, several limitations in the study need to be addressed in future research. For example, incorporating negative control bacteria, such as *Escherichia coli* or *Bacillus subtilis*, would strengthen the reliability of our findings. Additionally, we found co-infection of *M. abscessus* and *P. aeruginosa* in tuberculosis patients. Given that these bacteria can also co-infect individuals without tuberculosis, future studies should investigate whether tuberculosis influences the interaction between these pathogens. Furthermore, incorporating in vivo experiments would provide a more comprehensive and in-depth understanding of the underlying mechanisms. To increase the practical applicability of the study, the inclusion of diagnostic methods and molecular analyses would further enhance the significance of the research.

Given the stability of pathogen MVs and their ability to carry various potential clinical infection biomarkers [56,57], studies on the use of bacterial MVs for diagnostic purposes have already been reported. MVs can carry RNA molecules from bacteria [58], which can be utilized for the analysis of pathogen-specific transcripts. RNA analysis provides valuable insights into the active pathways within bacteria, such as those involved in immune evasion, antibiotic resistance, and virulence. Additionally, MVs contain lipids and metabolites that are part of bacterial metabolic processes [59], which can serve as biomarkers for acute or chronic infection detections. In future studies, we will focus on the characterization of MVs derived from co-infections with *M. abscessus* and *P. aeruginosa*. The objective is to identify specific biomarkers that enable rapid, highly sensitive, and early detection of *M. abscessus* and *P. aeruginosa* co-infections. The utilization of MVs for diagnosing these infections offers several notable advantages, including non-invasive sample collection (such as the straightforward isolation of MVs from patient samples, including sputum or plasma), rapid diagnostic potential, enhanced sensitivity, and the prospect for early detection.

## 5. Conclusions

Overall, our study primarily unveiled the intricate interaction between *P. aeruginosa* and *M. abscessus* in tuberculosis patients. We identified diverse strains of *M. abscessus* from clinical samples, indicating phenotypic variability, and demonstrated that *P. aeruginosa* inhibits the growth of *M. abscessus*, suggesting bacterial competition. Furthermore, the quorum-sensing molecule PQS secreted by *P. aeruginosa* enhances the production of MVs in *M. abscessus*, which may exacerbate cytotoxicity and immune toxicity. This study identified PQS as an inducer molecule of MV secretion by *M. abscessus*. Additionally, we verified that the biosynthesis of iron uptaking and the activation of the type VII secretion system may indirectly contribute to the regulation of increased MV secretion induced by PQS. However, the specific mechanisms require further investigation in the future. Our findings emphasize the critical importance of understanding the co-infection between *P. aeruginosa* and *M. abscessus* in infectious diseases, offering valuable insights into potential therapeutic strategies targeting PQS-mediated mechanisms in *M. abscessus* infections.

## Figures and Tables

**Figure 1 microorganisms-13-00116-f001:**
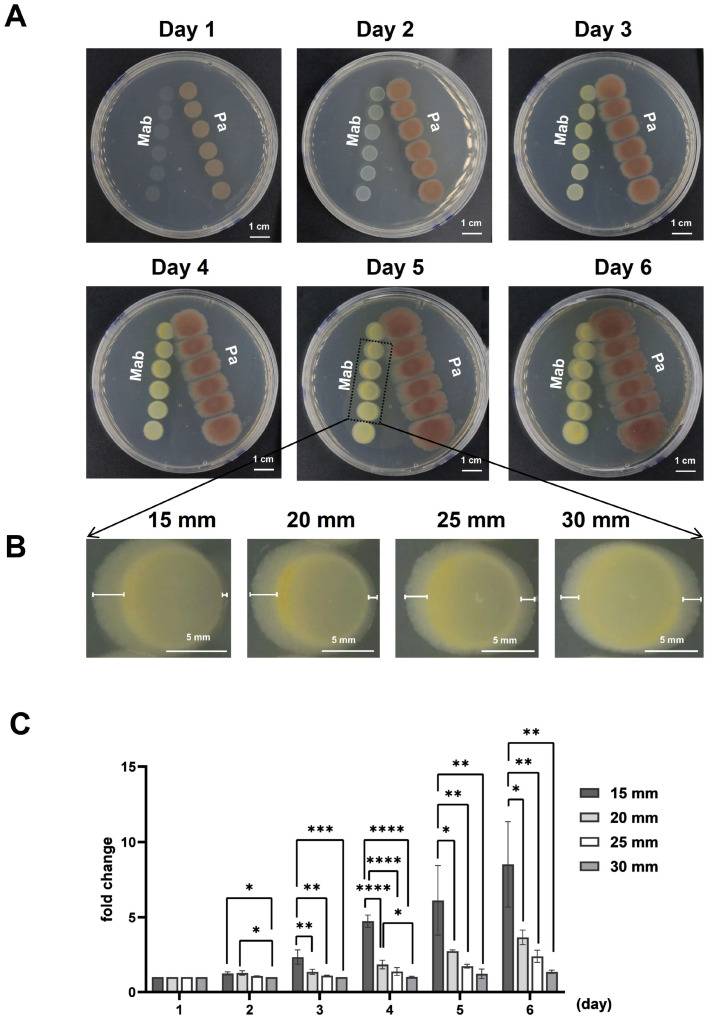
Interaction between *M. abscessus* and *P. aeruginosa*. (**A**) Pairwise proximity assays showing the interaction between *M. abscessus* 19977 (Mab) and *P. aeruginosa* PAO1 (Pa) at varying distances (15 mm, 20 mm, 25 mm, 30 mm). (**B**) Magnified view of *M. abscessus* colonies with the measurement method indicated. (**C**) Quantitative analysis of *M. abscessus* displacement in relation to the distance from *P. aeruginosa*. Data are expressed as Mean ± SD (n = 3). * *p* < 0.05, ** *p* < 0.01, *** *p* < 0.001, **** *p* < 0.0001.

**Figure 2 microorganisms-13-00116-f002:**
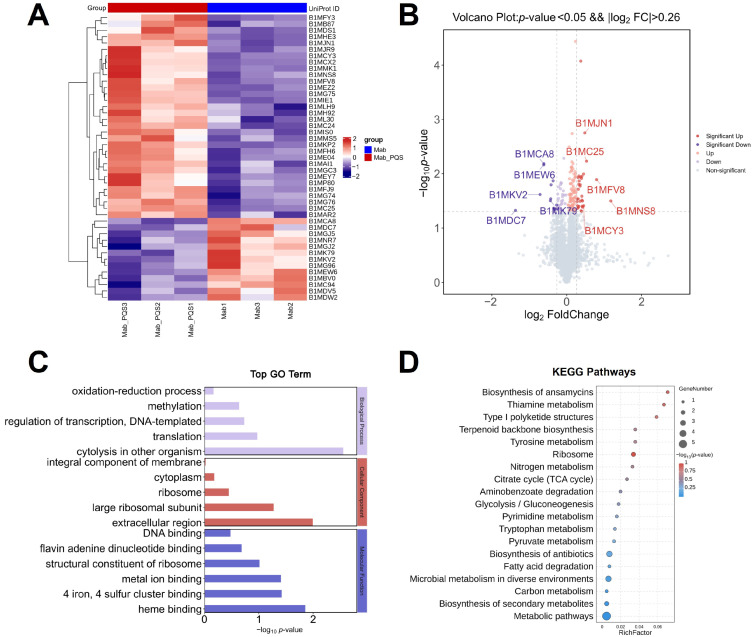
Proteomic analyses of *M. abscessus* after PQS treatment. The heatmap (**A**) and the volcano plot (**B**) of significantly differentially expressed proteins in the *M. abscessus* 19977 strain (Mab) + PQS proteome data compared to the Mab group. The top five upregulated and five downregulated DEPs are indicated in the volcano plot. (**C**) Gene Ontology (GO) annotation classification analysis of DEPs. (**D**) KEGG pathway analysis of DEPs in Mab + PQS compared to Mab group.

**Figure 3 microorganisms-13-00116-f003:**
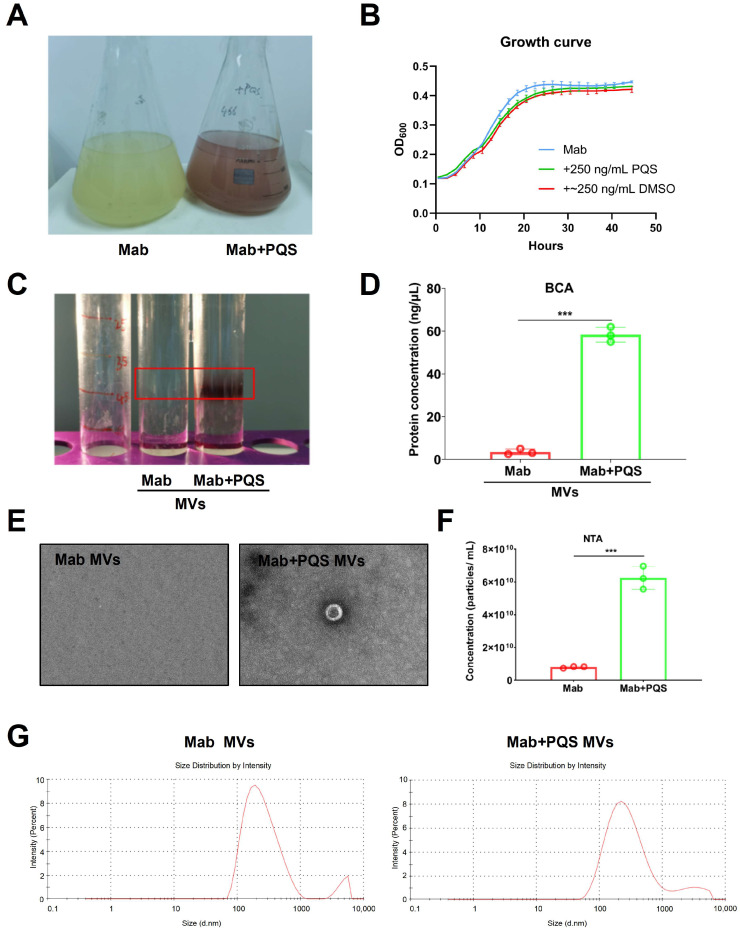
PQS increased MV productions in *M. abscessus* 19977 strain. (**A**) The image of the appearance of *M. abscessus* 19977 strain and Mab + PQS liquid cultures. (**B**) Growth curves of *M. abscessus* 19977 strain with 250 ng/mL DMSO and PQS, respectively. (**C**) Precipitation image of MVs via centrifugation separation. (**D**) Quantification of protein concentration in MVs. (**E**) Electron microscopy images of MVs in Mab and Mab + PQS groups. (**F**) MV concentration was measured by NTA in Mab and Mab + PQS groups (n = 3, Mann–Whitney *U* test). *** *p* < 0.001 compared with the Mab group. (**G**) MV particle size distribution measured by NTA in Mab and Mab + PQS groups. Data are expressed as mean ± SD (n = 3, Mann–Whitney *U* test). *** *p* < 0.001 compared with the Mab group. Data are expressed as mean ± SD.

**Figure 4 microorganisms-13-00116-f004:**
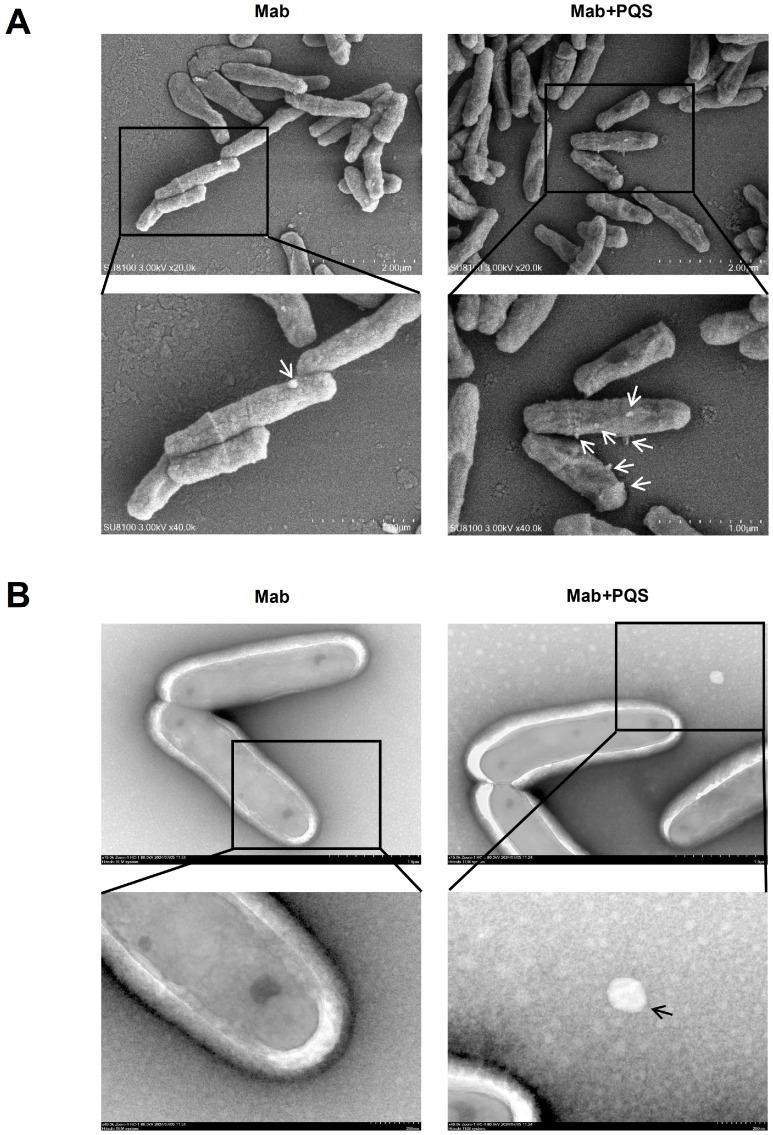
The impact of PQS exposure on the secretion of MVs and morphological changes in *M. abscessus*. (**A**) SEM images showing the surface of *M. abscessus* cells with and without exposure to PQS. MVs are indicated with arrows and the lower panels show the solid squares at higher magnification. (**B**) TEM images, following negative staining, illustrate the presence of MVs around *M. abscessus* cells. The lower panels show the solid squares at higher magnification.

**Figure 5 microorganisms-13-00116-f005:**
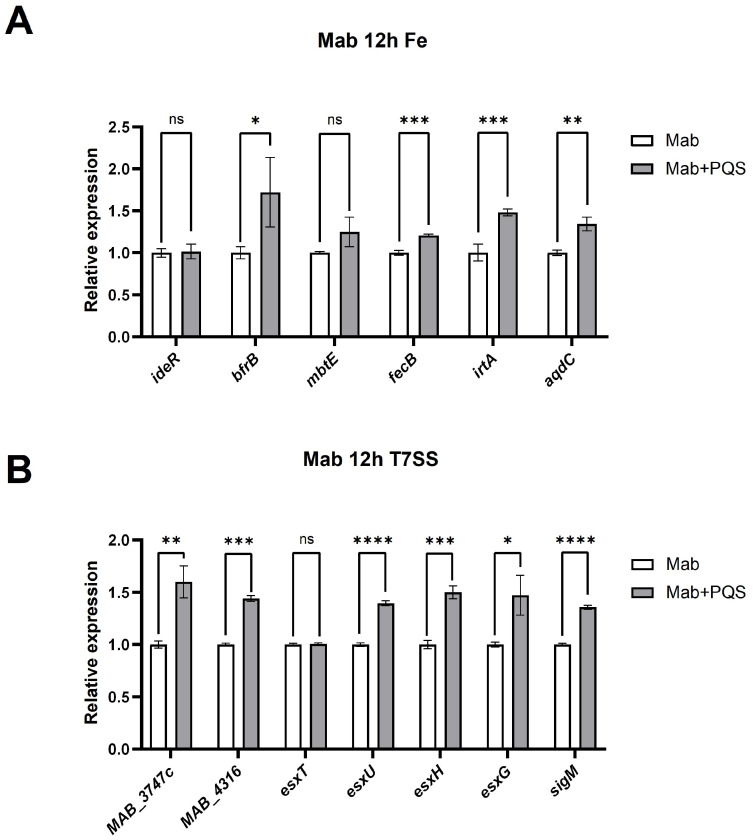
Quantitative real-time PCR (qPCR) analysis of select genes to verify proteomic results. (**A**) Relative mRNA expressions of the iron acquisition-related genes in PQS-treated *M. abscessus*. (**B**) Relative mRNA expressions of the type VII secretion system-related genes in PQS-treated *M. abscessus*. Values are expressed as relative expression with respect to the endogenous control gene, *sigA*. Data are expressed as Mean ± SD (n = 3). ns: not significant, * *p* < 0.05, ** *p* < 0.01, *** *p* < 0.001, **** *p* < 0.0001.

**Table 1 microorganisms-13-00116-t001:** Primer sequences of *M. abscessus*. F: Forward primers; R: reverse primers.

Targeted Gene	Genome Identifier	Primer Sequence (5′-3′)
*sigA*	MAB_3009-F	AGCGTGAGCTGCTACAGGAC
	MAB_3009-R	TGGATTTCCAGCACCTTCTC
*ideR*	MAB_3029-F	CTGGTCGACATCATCGGGTT
	MAB_3029-R	GATTGTTGAGCACCGTGAGC
*bfrB*	MAB_0127c-F	GGGATCGTGATGGCTACGTT
	MAB_0127c-R	TGCACGATCAAATCCGCAAC
*mbtE*	MAB_2248c-F	CCGATTCGTCGGTACGTCAT
	MAB_2248c-R	GCCATATCGCTTGCAGTTGG
*fecB*	MAB_4390-F	TCCGCACATACCGGTTTCAA
	MAB_4390-R	CCCAAGTCCGGAGAGGAAAC
*irtA*	MAB_2262c-F	TGAGTTTCTTGACCGACCCG
	MAB_2262c-R	TTCGTCTTTCTGGTGCTGCT
*aqdC*	MAB_0303-F	ATTGACGAGGCCTACCGAAC
	MAB_0303-R	CCCCGATCTTGGGATGTGAG
*MAB_3747c*	MAB_3747c-F	ACTTCGCGTAATCCGCTTCT
	MAB_3747c-R	ATTGTCTTTCGGTGGGGCAT
*MAB_4316*	MAB_4316-F	AGTCAGTCCTGAGTTGCTGC
	MAB_4316-R	TCGTTCTCAACGTGCTTGGA
*esxT*	MAB_3753c-F	CGCTGGTTGCTGATGTCAAG
	MAB_3753c-R	GTACTCCTGGTACGCAGTGG
*esxU*	MAB_3754c-F	CATCTCCTGGTCGAAGCGAG
	MAB_3754c-R	TTGCTCGATTCCACTGCCAA
*esxH*	MAB_2228c-F	AAGGTGTTGGTTTCGTGGGT
	MAB_2228c-R	CGGTGATACCTCGATGAGCC
*esxG*	MAB_2229c-F	GTCGAACGCATCAACGCC
	MAB_2229c-R	CTTGACGCACACATTCCCG
*sigM*	MAB_4938-F	GCTGTTTCGCCGTCATCATC
	MAB_4938-R	TCACCACGATGCGATAGAGC

**Table 2 microorganisms-13-00116-t002:** The top five differentially expressed proteins in the Mab + PQS group compared to the Mab group.

	Uniprot ID	Protein Name	Gene Name	Log_2_ (FoldChang)	*p*-Value
1	B1MC25	CBS domain-containing protein	*MAB_2717c*	0.530	5.892 × 10^−3^
2	B1MCY3	DUF3039 domain-containing protein	*MAB_3026c*	0.454	3.868 × 10^−2^
3	B1MFV8	Transcriptional regulator WhiB	*whiB*	0.797	1.284 × 10^−2^
4	B1MJN1	WXG100 family type VII secretion target	*MAB_4316*	0.482	1.776 × 10^−3^
5	B1MNS8	Uncharacterized protein	*MAB_1896c*	1.178	3.176 × 10^−2^
6	B1MCA8	MspA protein	*MAB_2800*	−0.613	6.532× 10^−3^
7	B1MDC7	1-deoxy-D-xylulose 5-phosphate reductoisomerase	*dxr*	−1.370	4.739 × 10^−2^
8	B1MEW6	Uncharacterized protein	*MAB_3496*	−0.615	6.869 × 10^−3^
9	B1MK79	Hypothetical porin	*MAB_1081*	−0.438	3.143 × 10^−2^
10	B1MKV2	Haemophore haem-binding domain-containing protein	*MAB_4528c*	−0.713	2.410 × 10^−2^

## Data Availability

The mass spectrometry proteomics data have been deposited in the ProteomeXchange Consortium via the iProX partner repository and can be accessed at https://www.iprox.cn/ (accessed on 5 January 2025), under the dataset identifier IPX0008842000.

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
