# Peer review of "Pseudomonas aeruginosa pqs Quorum Sensing Mediates Interaction with Mycobacterium abscessus In Vitro"

_microorganisms, 2025, doi:10.3390/microorganisms13010116_

Round 1
Reviewer 1 Report
Comments and Suggestions for Authors
Authors need to add in vitro to the title, as the observed interaction is very specific to the in vitro interactions. The experimental conditions lack a numerous feature of an in vivo coinfection.
There needs to be a proper introduction of co-occurrence of these organisms, complicating clinical outcomes. The individual epidemiology/burden/DALY needs to be covered in a introduction section. (https://pubmed.ncbi.nlm.nih.gov/27690688/) The quorum sensing section in intro belongs to discussion as was found in results and is not common knowledge. The intro should be about the problem and background. Also there is an increasing threat of multidrug-resistant strains of both P. aeruginosa and M. abscessus. The vulnerable patient population should also be described e.g.cystic fibrosis (CF) and other chronic lung diseases.
Currently, the intro is crashing right into the findings.
Authors need to ease in the reader by describing the background properly(https://www.biorxiv.org/content/10.1101/2024.01.22.576702v2
https://sciety.org/articles/activity/10.1101/2024.01.22.576702)
and disease management (https://respiratory-research.biomedcentral.com/articles/10.1186/s12931-023-02612-1)
Figure1 data significance is to be calculated
figure 2 has miniscule font
The discussion needs to be seriously toned down. There is no evidence that coinfections have these two agents growing in same proximity. These pathways could be due to nutrient depreviation.
Authors did not use any negative control bacteria. e.g. E coli/ B subtilis. Authors need to place a dedicated limitations paragraph and describe the shortcomings of the experiment setup
Author Response
Dear Reviewer,
Please kindly see the attachment.

Reviewer 2 Report
Comments and Suggestions for Authors
This is a very well structured study concerning tuberculosis co-infection with Pseudomonas aeruginosa. Co-infections are indeed probably the most important factor causing illness, much more single infections. The study presents useful insights, investigating the reasons of illness at all possible levels, microbiological, molecular and biochemical. I have only some minor modifications that would improve the quality of the manuscript:
I understand that the study is focused on human tuberculosis, but the disease is a common and maybe bigger problem in farmed animals as well. I therefore suggest to enrich the Introduction and Discussion with some parts concerning animals too. Co-infections are largely observed occasionally in cattle for example and play an important role for local economy and public health as an operational disease.
More details concerning the real time PCR analysis are needed. Which genes were targeted and according to which scenario? What was the research scope of analyzing these genes. Please add this info in the Materials and Methods in section 2.7 and also transfer Table S1 from supplementary material to basic manuscript.
Finally, in the discussion a diagnostic proposal would also benefit the utility of the findings. What do the authors propose for diagnosis, molecular analyses?
Round 2
Reviewer 1 Report
Comments and Suggestions for Authors
Authors have incorporated all of my comments.
in vitro needs to be in italics. The abbreviations need to be defined at first usage instance